# A *SUPERMAN-like* Gene Controls the Locule Number of Tomato Fruit

**DOI:** 10.3390/plants12183341

**Published:** 2023-09-21

**Authors:** Mi Zhang, Enbai Zhou, Meng Li, Shenglan Tian, Han Xiao

**Affiliations:** 1University of Chinese Academy of Sciences, 19A Yuquan Rd, Beijing 100049, China; zhangmi@cemps.ac.cn (M.Z.); zhouenbai@cemps.ac.cn (E.Z.); tianshenglan@sippe.ac.cn (S.T.); 2National Key Laboratory of Plant Molecular Genetics, CAS Center for Excellence in Molecular Plant Sciences, Institute of Plant Physiology and Ecology, Chinese Academy of Sciences, 300 Fenglin Rd., Shanghai 200032, China; limeng22387@126.com

**Keywords:** tomato (*Solanum lycopersicum*), *Solanum pimpinellifolium*, fruit morphology, flower development, locule number, *SUPERMAN*

## Abstract

Tomato (*Solanum lycopersicum*) fruits are derived from fertilized ovaries formed during flower development. Thus, fruit morphology is tightly linked to carpel number and identity. The *SUPERMAN* (*SUP*) gene is a key transcription repressor to define the stamen–carpel boundary and to control floral meristem determinacy. Despite *SUP* functions having been characterized in a few plant species, its functions have not yet been explored in tomato. In this study, we identified and characterized a *fascinated and multi-locule fruit* (*fmf*) mutant in *Solanum pimpinellifolium* background harboring a nonsense mutation in the coding sequence of a zinc finger gene orthologous to *SUP*. The *fmf* mutant produces supersex flowers containing increased numbers of stamens and carpels and sets malformed seedless fruits with complete flowers frequently formed on the distal end. *fmf* alleles in cultivated tomato background created by CRISPR-Cas9 showed similar floral and fruit phenotypes. Our results provide insight into the functional conservation and diversification of *SUP* members in different species. We also speculate the *FMF* gene may be a potential target for yield improvement in tomato by genetic engineering.

## 1. Introduction

Tomato is one of major vegetable fruit crops cultivated worldwide, providing a rich resource of nutrients for human diets. As a typical type of fleshy fruit, tomato fruits develop from fertilized ovaries undergoing substantial post-fertilization growth, but many features of fruit morphology are defined during flower development. For example, the elongated and pear fruit shapes found in some cultivars are mainly defined by ovary shape before fertilization. Indeed, the two major fruit shape genes *SUN* and *OVATE* set ovary shape [1,2]. In cultivated tomato, the fruits have multiple locules derived from carpels. Thus, the locule number is largely attributed to the number of carpels formed during flower development, which is mainly controlled by *LOCULE NUMBER* (*LC*, ortholog of *WUS*) and *FASCIATED* (*FAS*, ortholog of *CLV3*) [3,4,5,6].

The bisexual tomato flower, like most angiosperm flowers, consists of four distinctive whorls of floral organs: sepals in the outmost whorl, petals in the second whorl, and stamens and carpels in the respective third and fourth whorls. The number of individual floral organs are varied among cultivated tomato accessions, but the wild ancestor *S. pimpinellifolium* flower has five sepals, five petals, five stamens fused at the base to form a cone-like structure, and a single pistil formed by two fused carpels [7]. The number and position of floral organs in each whorl is governed by floral homeotic genes and cadastral genes controlling floral organ boundary [8,9]. *SUPERMAN* (*SUP*) is the first identified boundary gene playing a crucial role in maintaining the boundary between stamen and carpel whorls in Arabidopsis flowers because the typic floral phenotype observed in *sup* mutants is the increased number of stamens at the expense of carpels [10,11]. The extra stamens in *sup* mutants likely resulted from an identity change in some whorl 4 cells, through which their fate was changed from female to male [12]. *SUP* encodes a transcription factor containing a C_2_H_2_ type zinc finger DNA binding domain and an EAR-like repressor domain [11,13,14]. Its specific expression at the boundary between whorl 3 and 4 prevents the expression of *APETALA3* (*AP3*) and *PISTILLATA* (*PI*) reaching the central region of the floral meristem [11,12].

Characterization of *SUP* orthologs in a few plant species, including rice (*Oryza sativa*), *Petunia hybrida,* and *Medicago truncatula,* has revealed functional conservation and divergence. The *sup* mutants of *Petunia hybrida* (*phsup1*) also show an increased stamen number at the expense of carpels, but form extra tissues connecting the inner three whorls [15]. Rice *small reproductive organs* (*sro*) harboring a mutation in a *SUP-like* gene does not show defects in floral organ number and organ identity, instead the mutant has smaller stamens and pistils [16]. Mutations in the Medicago *SUPERMAN* (*MtSUP*) gene cause an increased number of the inner three whorls and more flowers formed on individual inflorescence, indicating the *MtSUP* gains novel functions in the legume species [17]. It has been proposed that the transcriptional divergences detected in the flowers between Arabidopsis and Medicago underlie the functional diversification [18]. Similarly, this kind of expression divergence may be also applicable to the rice *SUP* ortholog, since *SRO* is specifically expressed in stamen filaments, not at the stamen–carpel boundary [16]. Because *SUP* orthologs have been only characterized in a few plant species, it remains to determine whether expression divergence explains the evolution of *SUP* functions. Nevertheless, the phenotypic variations observed in these *sup* mutants highlight the need to explore the functions of *SUP* orthologs in more plant species. 

As shape and size are important fruit quality traits in tomato [19,20,21,22], understanding how key regulatory genes control the number of floral organs, especially the carpel number, not only sheds new light on gene conservation and diversification but also provides new strategies for breeding new varieties to meet the increasing demand for diverse fruit quality. In this study, we report the characterization of *FMF*, orthologous to *SUP*, regulating floral organ number, and ovary development in tomato.

## 2. Results

### 2.1. Phenotypic Analysis of the Fmf Mutant

Through screening for mutants showing fruit phenotypes including shape and size, we identified one such mutant in an Ethyl Methane Sulphonate (EMS)-mutagenized S. pimpinellifolium (accession LA1781) population. The mutant named *fasciated and multi-locule fruit* (*fmf*) had abnormal fruit morphology. The *fmf* mutant also exhibited abnormal flower development. Compared with the wild type (LA1781), *fmf* flowers had apparently normal sepals and petals, but its stamen cones were cracked and styles often split and shorter (Figure 1A–H). Dissection of floral organs revealed that *fmf* flowers had more than five stamens, in contrast to the five stamens constantly observed in the wild type flower (Figure 1I,J). *fmf* carpels had either unfused multiple ovaries connected at the base or multiple styles failed to be fused together at the distal end (Figure 1K–P). fmf ovaries contained multiple locules (carpels) and in most cases without ovule developed (Figure 1Q–V). Quantification of floral organs at anthesis showed that the numbers of stamens and carpels had significantly increased in *fmf*, on average 8.9 stamens and 6.2 carpels were formed (Figure 1W). *fmf* styles were dramatically shortened (Figure 1X). In addition, *fmf* mutation had a weak impact on inflorescence development: *fmf* inflorescences were shorter, containing slightly fewer flowers (Supplemental Appendix A). 

Though *fmf* set fruits poorly (Supplemental Appendix A), multiple-locule parthenocarpic fruits were formed occasionally (Figure 2A–D). Strikingly, complete flowers were often formed at the distal ends of these seedless *fmf* fruits (Figure 2E). To test the functionality of *fmf* stamens and carpels, we either pollinated *fmf* pistils with wild type pollens or wild type pistils with *fmf* pollens. *fmf* flowers pollinated with wild type pollens produced seedless fruits, whereas wild type plants produced seeded fruits when pollinated with *fmf* pollens (Figure 2F,G). These results suggest that *fmf* is female-sterile but male-fertile.

We further investigated the development of *fmf* flowers by histological analysis and scanning electronic microscopy (SEM). Histological analysis revealed that the *fmf* mutation did not affect sepal and petal formation (Figure 3A–E). The first noticeable difference between wild type and *fmf* flowers was observed after stamen primordia were formed, which meant that in *fmf* flowers an additional whorl of stamen primordia was developed in between the third and fourth whorls (Figure 3C,F). Later, *fmf* flowers developed more carpels (Figure 3G–L). In the center of the *fmf* flowers, there were often flower structures (Figure 3J). SEM analysis also confirmed that the timing and position of sepal and petal primordia were not impacted by *fmf* mutation (Figure 4A–D), while extra stamen and carpel primordia were developed afterward (Figure 4E–H). When wild type flowers reached anthesis stage, their pistils were almost closed, like a closed mouth (Figure 4I,J). In contrast, the pistils failed to be enclosed in *fmf* flowers (Figure 4K,L). Within the *fmf* flowers, new complete flower-like structures were also observed (Figure 4K). These results indicate that mutation in the *FMF* gene mainly affects the development of the two innermost whorls, stamens, and carpels.

### 2.2. Molecular Cloning of the FMF Gene

To identify the causal mutation underlying the *fmf* phenotypes, we generated an F_2_ population by crossing the *fmf* mutant to cv. Moneymaker (LA2706). Rough mapping was conducted by BSA-seq using pooled genomic DNA isolated from wild type and *fmf* plants. The *FMF* locus was placed on a region around 66 Mb on chromosome 9 (ITAG4.0) showing maximal difference in the SNP index (Figure 5A). Then, we performed fine mapping using 384 *fmf* plants from the segregation population. Using Indel markers, the *FMF* locus was further narrowed down to an 83.85 kb interval between markers xps2515 and xps2452. The interval contains four annotated genes: *Solyc09g089580*, *Solyc09g089590*, *Solyc09g089600*, and *Solyc09g089610*, which encode 2-oxoglutarate (2OG) and the Fe(II)-dependent oxygenase superfamily protein, transcriptional regulator TAC1, the zinc finger protein, and the ethylene receptor-like protein (ETR6), respectively (Figure 5B). After sequencing the four candidate genes, the *fmf* mutant only contains a nonsense mutation in *Solyc09g089590*, which the cytosine at position 306 of the coding sequence was changed to adenine and the mutation introduced a stop codon after translation of 101 amino acids (102S*). Solyc09g089590 is a putative C_2_H_2_-type transcription factor containing transcription repression domain—the EAR-motif [13,14]. The deduced fmf protein was truncated, lacking the C-terminal containing the EAR-motif, suggesting that the *fmf* mutation may impair its transcription in regulation activity. 

Sequence and phylogenetic analysis of Solyc09g089590 and its close homologs from tomato and other species showed that Solyc09g089590 shares the highest similarity (e-value = 2 × 10^−24^) and is grouped with Arabidopsis SUP and Petunia PhSUP1. SUP is thought to function as a cadastral gene to maintain the floral organ boundaries, in which its mutation causes formation of extra stamens at the expense of carpels [10]. Given *fmf* shows similar stamen phenotype with Arabidopsis *sup* mutants, Solyc09g089590 is likely orthologous to SUP and the missense mutation in this gene is responsible for the observed *fmf* floral morphology. However, tomato likely have a paralog Solyc06g053720 close to SUP; it also shares the highest sequence identity with SUP (e-value = 2 × 10^−25^). Moreover, on the inferred phylogenetic tree, Solyc09g089590/FMF was not grouped with Arabidopsis TELOMERASE ACTIVATOR1 (TAC1), suggesting that it is unlikely orthologous to TAC1 as annotated by international tomato genome sequencing project (version ITAG4.0). 

### 2.3. Creation of New fmf Alleles by CRISPR-Cas9

To further confirm that *Solyc09g089590* underlies the *FMF* locus, we generated *Solyc09g089590* mutants in Moneymaker background by CRISPR-Cas9. We obtained two different alleles *fmf-cr2* and *fmf-cr4* that showed *Solyc09g089590* was successfully edited; *fmf-cr2* had 4 bp deletion (+182~+185 start from start codon ATG) and *fmf-cr4* had 9 bp deletion (+175~+183) (Supplemental Appendix A). The deletion in *fmf-cr2* introduced premature translation stop codon after 63 amino acids, which disrupted the conserved C_2_H_2_ zinc finger DNA binding domain encoding by *Solyc09g089590*. The *fmf-cr4* mutation resulted in loss of the first three amino acids of the invariant QALGGH motif in the C_2_H_2_ domain, which also likely disrupted this functional domain. *fmf-cr2* and *fmf-cr4* showed identical flower phenotype (Supplemental Appendix A–D, Figure 6). Thus, a more detailed phenotypic analysis was conducted just on the *fmf-cr2* allele. Like the *fmf* mutant in LA1781 background, *fmf-cr2* in Moneymaker background also had increased numbers of stamens and carpels, shorter and unfused styles (Figure 6A–S). *fmf-cr2* was also male fertile and set seedless fruits, but unlike *fmf* fruits in LA1781 background, *fmf-cr2* fruits only had cracks, no extra flower structure was formed in the fruits (Figure 6T–X), suggesting that mutations in the *FMF* gene have different impacts on cell proliferation and differentiation in whorl 4 between cultivated tomato and its wild relatives. Nevertheless, both in LA1781 and Moneymaker backgrounds, mutations in the *FMF* gene caused very similar defects in stamen and carpel development. 

### 2.4. FMF Expression during Flower Development

We investigated *FMF* expression in developing flowers of wild type by in situ hybridization. *FMF* transcripts were detected in stamen and carpel primordia at the early flower stage, then in stamens when all floral organs were formed (Figure 7A–D). FMF was also expressed weakly in petals and carpels, but not in sepals. Because *WUS* is a crucial regulator of floral meristem identity and its ortholog *SlWUS* controls locule number in tomato [3,6,23], we also compared its expression in wild type and *fmf* flowers. Compared with wild type, *SlWUS* expression was much stronger in stamen and carpel primordia of *fmf* flowers (Figure 7E,F). 

## 3. Discussion

In many cases, the fruit develops from the ovary after fertilization. Thus, flower development has considerable impact on fruit morphology. For example, *SUN*, *OVATE*, *LC/SlWUS*, and *FAS/SlCLV3* control tomato fruit shape and size through their early actions on ovary development [1,2,3,4,6,23,24]. In this study, we identified *FMF* that is likely orthologous to the Arabidopsis *SUP* gene controlling tomato fruit morphology. The *fmf* mutant identified in EMS-mutagenized *S. pimpinellifolium* LA1781populations and the *fmf-cr* mutants in *S. lycopersicum* cv. Moneymaker created by CRISPR-Cas9 showed almost identical flower and fruit phenotypes except no flower structures were observed in the *fmf-cr* fruits (Figure 1, Figure 2 and Figure 6). All *fmf* mutants contained more stamens and carpels, shorter and often unfused styles, and fewer if not absent ovules. Moreover, the *fmf* mutants produced functional pollens. These morphological features make the *FMF* gene a potential target for genetic manipulation to create desirable fruit traits in tomato and other Solanaceous crops, i.e., creating weak *sup* alleles to increase locule number. 

Different *sup* alleles exhibit a wide spectrum of flower abnormality in Arabidopsis and *M. truncatula*, ranging from superman to superwoman to supersex [8,17,18,25,26,27,28]. Compared with the *sup* mutants identified in other plant species, *fmf* mutants only had supersex phenotype-producing bisexual flowers with more stamens and carpels, sharing higher similarity with Arabidopsis *sup-5* and *M. truncatula mtsup-2* [8,17]. No *fmf* flower showed superman (having extra stamens and carpelloid carpels) or superwoman (bisexual flowers with all stamens on a single whorl and an indeterminate whorl 4) morphology as observed in some alleles of Arabidopsis and *M. truncatula sup* mutants. Similarly, low spectrum of phenotypic variations was also detected in the rice *sro* mutant and the *sup* mutants of *P. hybrida* [15,16]. The differences in phenotypic variations observed in allelic *sup* mutants across plant species imply that *SUP* regulation may be integrated into species–specific genetic networks controlling floral development. 

Characterization of *sup* mutants in several plant species including Arabidopsis, *M. truncatula*, *P. hybrida*, and rice has revealed the conservation and diversification of *SUP* functions across plant species [8,10,11,12,15,16,17,25,27,28,29,30,31,32,33,34,35,36,37,38,39,40,41,42]. *fmf*, *sup*, *phsup1* and *mtsup* mutants have increased numbers of stamens and/or carpels in their respective flowers. Therefore, the four eudicot species have conserved *SUP* functions in regulation of floral organ numbers. However, rice *rso* flowers have reduced-size stamens and carpels but no changes in the number and identity of floral organs [16], suggesting that *SUP* functions are not well conserved between eudicots and monocots. It is also possible that SRO is not the true ortholog of Arabidopsis SUP because it is grouped with RABBIT EAR (RBE), a close homolog of SUP, though their relationship is not well supported by phylogenetic analysis. SUP belongs to a subclade of the zinc finger protein family, containing a single C_2_H_2_ zinc finger DNA binding domain and an EAR-repressor domain. It is notable that members in this subclade share low sequence similarity in the regions beyond the two domains as indicated by their phylogenetic relationships (Figure 5C). However, FMF is closer to Petunia PhSUP1 and Arabidopsis SUP, providing additional evidence to support that FMF is orthologous to SUP. 

The differences in flower phenotypes among *sup* mutants of the five species may be explained by different spatiotemporal expression patterns of *SUP* orthologs. *FMF* is expressed in whorl 2 and 3 at early stage and mainly expressed in stamens at late stage. *SUP* is expressed on the two sides of the stamen–carpel boundary [10,11,12,27,31,32]. However, such an expression pattern is not observed for *SRO*, *MtSUP,* and *PhSUP1* [15,16,17]. For example, *MtSUP* is expressed in carpel primordia and the common primordia where petals and stamens developed later, and *SRO* is specifically expressed in stamen filaments. 

*fmf* mutants have a distinctive floral phenotype not observed in any other *sup* mutants: complete flowers formed in the *fmf* ovaries/fruits in the genetic background of *S. pimpinellifolium* LA1781 (Figure 2 and Figure 6). Such a phenotype indicates that floral meristem is not terminated in a timely manner in *fmf* flowers. It remains to further explore whether additional genetic components are involved and what caused the incomplete penetration in the cultivar Moneymaker. Given *WUS* is required for floral meristem identity [43], we speculate that the *SlWUS* activity in LA1781 and Moneymaker flowers may be somewhat different because the increased locule number in cultivated tomato is highly associated with DNA variations in the *SlWUS* gene. In addition, *FMF* mutations also slightly affected inflorescence length and the number of flowers formed on individual inflorescence. Nevertheless, our results from characterization of the *FMF* gene in tomato reveal an undescribed *SUP* function. 

## 4. Materials and Methods

### 4.1. Materials and Plant Growth Conditions

*S. pimpinellifolium* accession LA1781 and cultivated tomato cv. Moneymaker (LA2706) were grown in phytotrons or plastic greenhouse. When grown in phytotron, plant pots in blonde peat (Pindstrup Mosebrug A/S, Ryomgaard, Denmark) were grown under the condition of temperatures of 18 to 25 °C with a relative humidity of 70–80% and illuminated by 150 mE·m^−2^·s^−1^ light intensity for 16 h. Plants grown in a plastic greenhouse were under natural solar radiation, and the during the growth season (late March to middle of July) the temperature ranged from 15 to 29 °C and relative humidity from 32 to 90%.

### 4.2. Bulked-Segregant Analysis-Seq (BSA-Seq) and Map-Based Cloning

After examination of fruit morphology, individual *fmf* plants segregating in a F2 population derived from a cross between *fmf* (pollen donor) and Moneymaker were sampled for DNA extraction using method previously described [2]. 

For BSA-seq, additional sampling was conducted, which leaves from around 100 plants for each genotype were pooled for the mutant and wild type before DNA extraction. The two pooled DNA samples were sequenced by Hiseq 2500 (Illumina). Raw reads were quality checked by *fastqc* (version v0.11.9, http://www.bioinformatics.babraham.ac.uk/projects/fastqc/, accessed on 7 September 2019), followed by the removal of low-quality reads and adaptor sequences by *Trimmomatic* (version 0.39, accessed on 22 February 2021) [44]. The processed clean reads were mapped to the tomato reference genome ITAG4.0 downloaded from Sol Genomics Network (https://solgenomics.net/, accessed on 22 February 2021) using BWA-MEM (version BWA-0.7.17, 15 August 2021) [45]. Mapping results were sorted by *samtools faidx* (version 1.13, accessed on 9 August 2021) [46]. We used the functions AddOrReplaceReadGroups, MarkDuplicates, BamIndexStats in the Picard software (release 1.119. http://broadinstitute.github.io/picard/, accessed on 16 January 2022) to further remove duplicates and index the mapped reads. SNPs were called using the HaplotypeCaller program in *GATK* (v4.2.4.0, accessed on 16 January 2022) with following parameters: -stand-call-conf 30 --native-pair-hmm-threads 15 -mbq 20 [47]. The SNP index was then calculated for each allele as described previously [48]. Sliding window analysis on SNP-index plots was carried out with 1 Mb window size and 10 kb increment and the plot graph was draw by *ggplot2* (https://github.com/tidyverse/ggplot2, accessed on 20 February 2022). 

After rough mapping, fine mapping was focused on an 8.14 Mb region (60.28–68.42 Mb) using developed Indel and CAPS markers. Marker information can be found in Appendix A.

### 4.3. Generation of fmf-cr Mutants by CRISPR-Cas9

The design of target oligoes and vector construction were previously described [48]. Briefly, target oligoes containing AAATCAGCTCAAGCTCTTGG (start at 166 after ATG) were designed using the online tool CRISPR-P (http://cbi.hzau.edu.cn/crispr/, accessed on 6 March 2022) [49]. Oligoes after annealing were cloned into the psgR-Cas9-At vector and further assembled into pCambia1300 [50]. The agrobacterium tumefaciens strain GV3101 harboring the plasmid was used for plant transformation using Moneymaker cotyledons as explants according to the method previously described [51]. 

### 4.4. Phylogenetic Analysis of FMF Homologs

Homologous proteins of Arabidopsis SUP and tomato FMF were identified by BlastP using FMF protein sequence as query on Araport11 protein sequences of Arabidopsis (TAIR, https://www.arabidopsis.org/, accessed on 7 December 2022) and ITAG4.0 protein sequences of tomato (SGN, https://solgenomics.net/, accessed on 7 December 2022). The sequences of SUP orthologs in Petunia, rice, and Medicago were retrieved from NCBI (https://www.ncbi.nlm.nih.gov/protein/, accessed on 7 December 2022). Phylogenetic analysis was conducted using the MEGA7.0 software [52]. The original phylogenetic tree was constructed using the neighbor-joining method and then the bootstrap consensus tree was inferred from 1000 replicates. 

### 4.5. Microscopy

Flower buds at different developmental stages were collected and fixed in FAA fixative solution at 4 ℃ overnight. The samples were dehydrated through ethanol series (50% –70%–85%–95%–100%). For histological analysis, the flower samples were embedded in Paraplast^®^ (P3558, Sigma-Aldrich, St. Louis, MO, USA) and 10 μm sections were made by a Leica microtome using a Leica microtome (Leica) and were briefly stained by 0.05% toluidine blue. For SEM analysis, the flower samples after dehydration were subjected to critical point drying in liquid nitrogen and coated with gold, then were examined under an electronic microscope (Zeiss Merlin Compact, Oberkochen, Germany). 

### 4.6. In Situ Hybridization

Flower buds were fixed in FAA overnight at 4 ℃ and embedded in Paraplast (Sigma-Aldrich, St. Louis, MO, USA) after dehydration. The samples were then sliced into 8 μm sections using a microtome (Leica, Wetzlar, Germany). After dewaxing and rehydration, the sections were probed by digoxigenin-labeled sense and antisense riboprobes as previously described [53].

## Figures and Tables

**Figure 1 plants-12-03341-f001:**
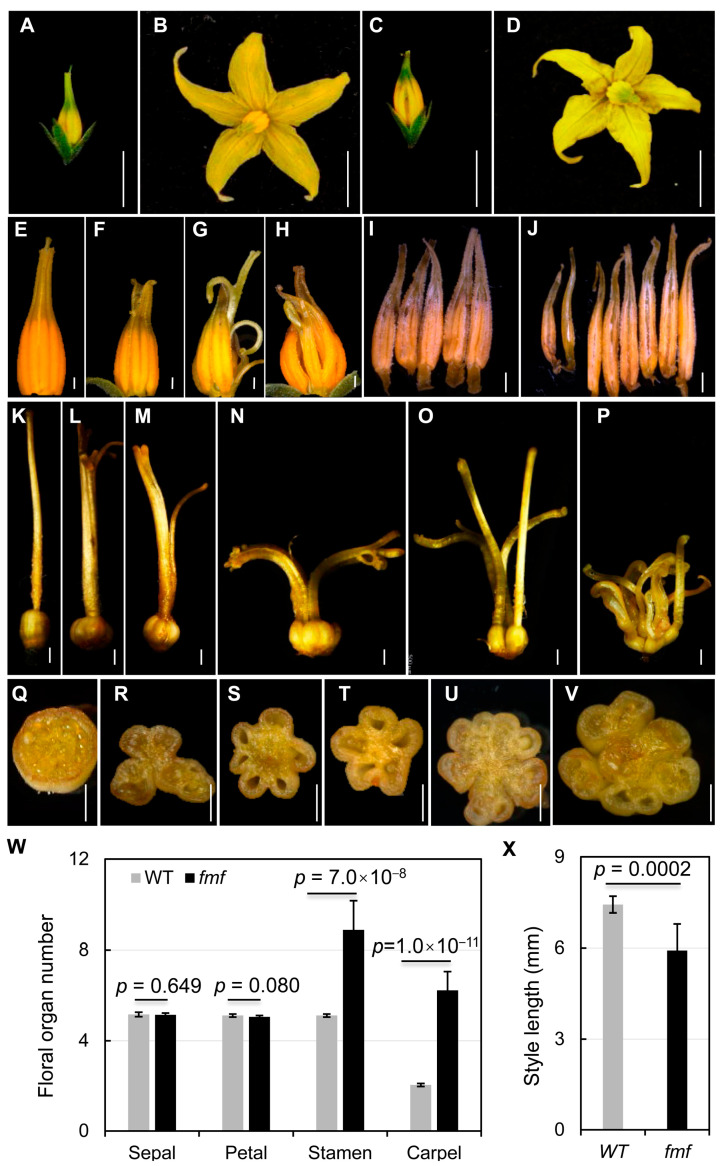
Floral phenotypes of *fmf* and wild type. (**A**–**D**) Unopened (**A**,**C**) and anthesis (**B**,**D**) flowers of the *fmf* mutant (**C,D**) and wild type ((**A**,**B**), LA1781). (**E**–**H**) Anther cones of wild type (**E**) and *fmf* (**F**–**H**). (**I**,**J**) Dissected anthers from a single flower of wild type (**I**) or *fmf* (**J**). (**K**–**P**) Dissected carpels of wild type (**K**) and *fmf* (**L**–**P**). (**Q**–**V**) Transversely spliced ovaries of wild type (**Q**) and *fmf* (**R**–**V**). (**W**) Quantification of the numbers of floral organs of wild type and *fmf*. (**X**) Style length of wild type and *fmf*. Measurements in (**W**,**X**) were conducted on 100 flowers from 10 plants for each genotype. Data represent means ± SD. *p* values were calculated using Student’s *t*-test. Scale bar: 5 mm (**A**–**D**), 500 mm (**E**–**V**).

**Figure 2 plants-12-03341-f002:**
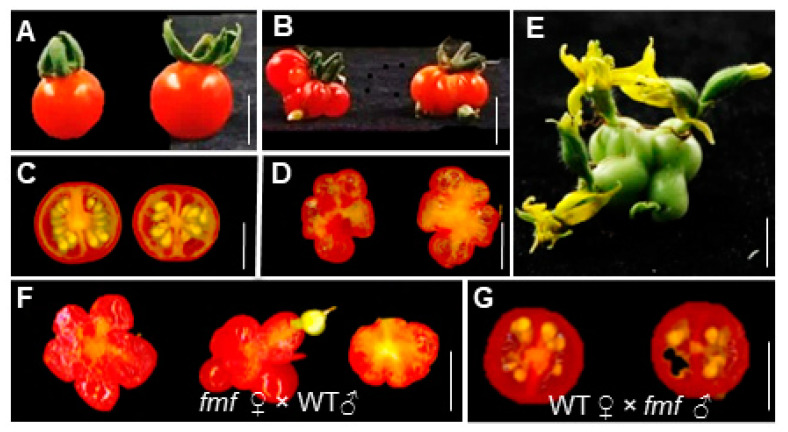
Fruit development of *fmf* and wild type. (**A**,**B**) Representative images of mature fruits of wild type (**A**) and *fmf* (**B**). (**C**,**D**) Transversely sliced fruits of wild type (**C**) and *fmf* (**D**). (**E**) A representative *fmf* fruit showing extra flowers developed on the distal parts of the fruit. (**F**) Images of parthenocarpic fruits developed from *fmf* flowers pollinated with wild type pollens. (**G**) Seeded fruits developed from wild type flowers pollinated with *fmf* pollens. Scale bar, 1 cm.

**Figure 3 plants-12-03341-f003:**
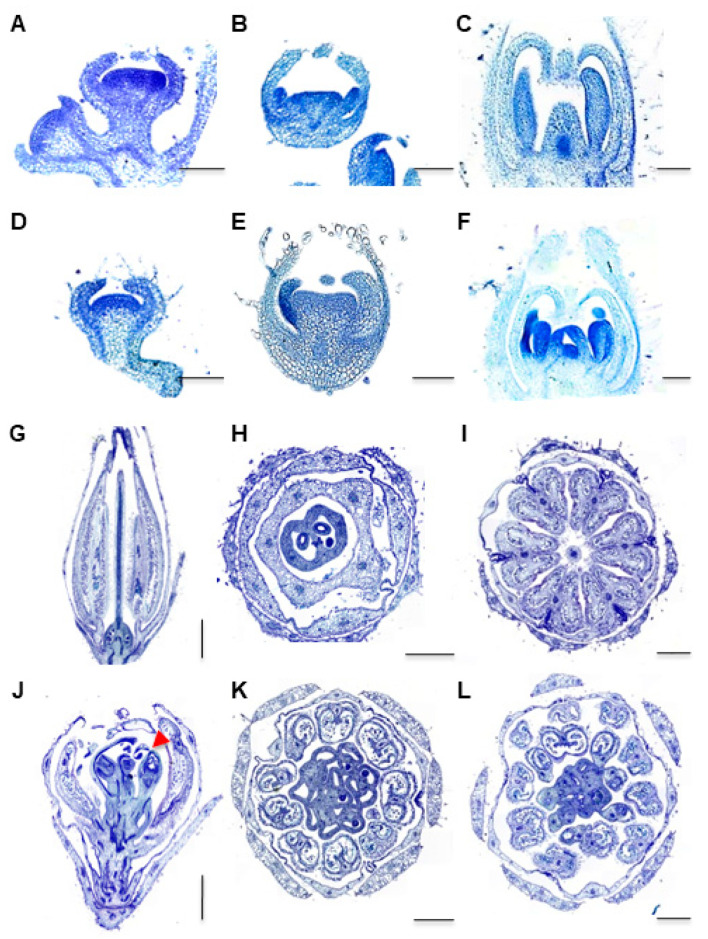
Histological analysis of *fmf* and wild type flowers. (**A**,**C**) Parafilm sections of type flower buds at the stages when sepal (**A**), petal (**B**), and stamen (**C**) was formed, respectively. (**D**–**F**) Parafilm sections of *fmf* flower buds at stages similar to wild type showing in (**A**–**C**). (**G**–**I**) Parafilm sections of mature wild type flowers (one day before anthesis). (**J**–**L**) Parafilm sections of mature *fmf* flowers. Longitudinal (**G**,**J**) and transverse (**H**,**I**,**K**,**L**) sections were made to reveal the structures of inner floral organs. The red arrowhead in (**J**) indicates a flower-like structure. Scale bar: 100 μm (**A**–**F**), 500 μm (**H**,**I**,**K**,**L**), 1 mm (**G**,**J**).

**Figure 4 plants-12-03341-f004:**
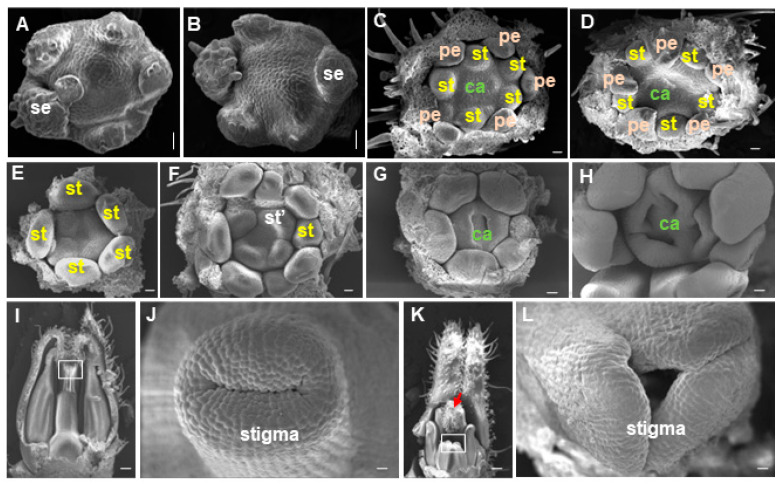
SEM analysis of *fmf* and wild type flowers. (**A**,**B**) Flower buds of wild type (**A**) and *fmf* (**B**) at sepal initiation stage. (**C**,**D**) Flower buds of wild type (**C**) and *fmf* (**D**) in which petal primordia initiated. (**E**,**F**) Second whorl of stamen primordia formed in *fmf* flower buds. (**G**,**H**) Carpel fusion just started in wild type (**G**) and *fmf* (**H**) flower buds. Multi-carpels were observed in *fmf*, contrasting to two carpels in wild type. (**I**–**L**) Overview of mature wild type (**I**,**K**) and *fmf* (**K**,**L**) flowers and close examination of their stigma (**J**,**L**). The white boxes in (**I**,**K**) indicate stigma parts observed in (**J**,**L**) and the red arrow in (**K**) points to a flower-like tissue developed within the *fmf* flower. se, sepal; pe, petal., st, stamen; ca, carpel; st’, extra stamen. Scale bar: 20 μm (**A**–**H**), 100 μm (**I**,**K**), 10 μm (**J**,**L**).

**Figure 5 plants-12-03341-f005:**
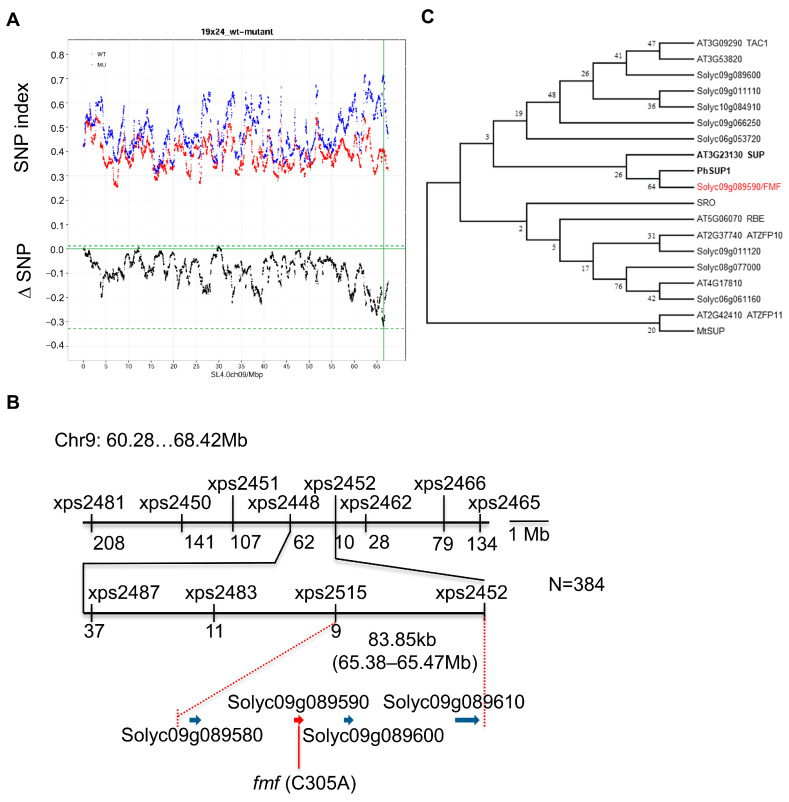
Cloning of the *FMF* gene. (**A**) SNP index of bulked DNA samples of wild type and *fmf* plants from a F2 population derived from a cross between *fmf* (in LA1781 background) and Moneymaker (LA2706). The graph shows that maximal SNP deviation between *fmf* and wild type was detected around 67 Mb on chromosome 9. (**B**) Fine mapping of the *FMF* locus. The numbers beneath the markers with the names starting with xps are the numbers of recombinants. *FMF* was placed on an interval of 83.85 kb region containing four predicted genes. (**C**) Phylogenetic tree of FMF and its close homologs in tomato and Arabidopsis. The tree was generated using MEGA7 with bootstrap of 1000.

**Figure 6 plants-12-03341-f006:**
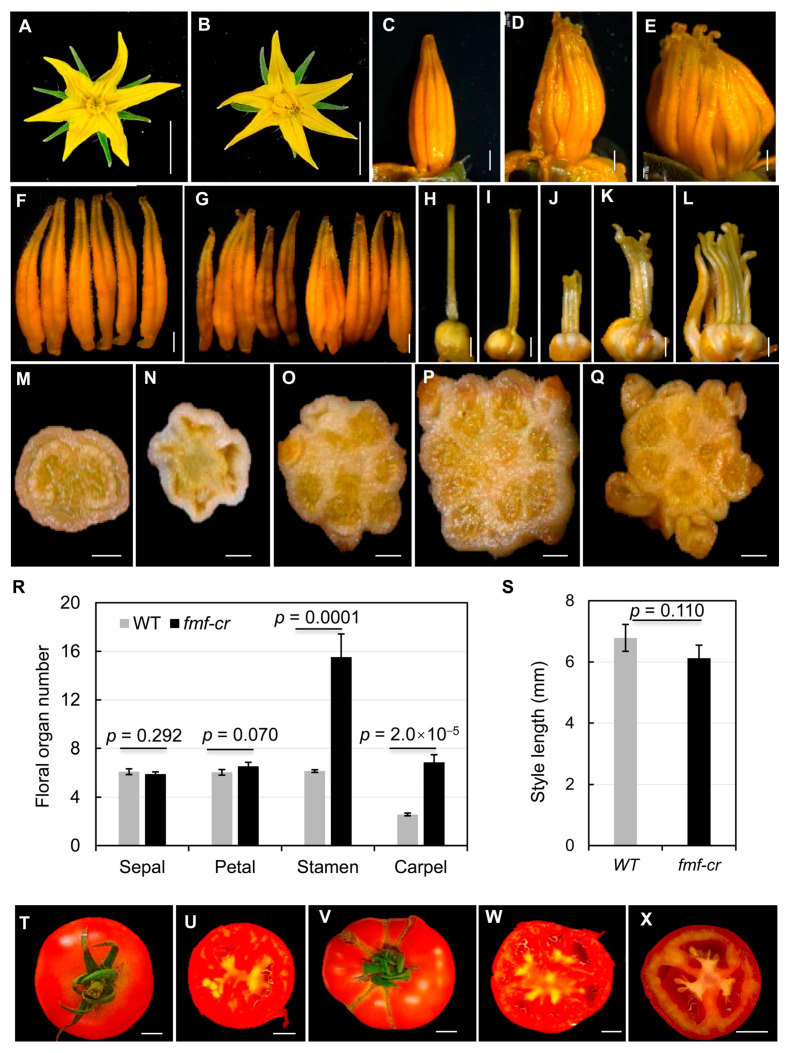
Flower and fruit phenotypes of *fmf-cr2* in cultivated tomato background. (**A**,**B**) Anthesis flowers of the *fmf-cr2* mutant (**B**) and wild type ((**A**) Moneymaker). (**C**–**E**) Anther cones of wild type (**C**) and *fmf-cr2* (**D**,**E**). (**F**,**G**) Dissected anthers from a single flower of wild type (**F**) or *fmf-cr2* (**G**). (**H**–**L**) Dissected carpels of wild type (**H**) and *fmf-cr2* (**I**–**L**). (**M**–**Q**) Transversely spliced ovaries of wild type (**M**) and *fmf* (**N**–**Q**). (**R**) Quantification of the numbers of floral organs of wild type and *fmf*. (**S**) Style length of wild type and *fmf*. Measurements in (**R**,**S**) were conducted on 40 flowers from 4 plants for each genotype. Data represent means ± SD. *p* values were calculated using Student’s *t*-test. (**T**–**W**) Mature fruits of wild type (**T**,**U**) and *fmf-cr2* (**V**,**W**). No seed was observed in *fmf-cr2* fruits. (**X**) Seeded fruits developed from wild type flowers pollinated with *fmf-cr2* pollens. Scale bar: 1 cm (**A**–**E**,**T**–**X**), 1 mm (**F**–**L**), 500 μm (**M**–**Q**).

**Figure 7 plants-12-03341-f007:**
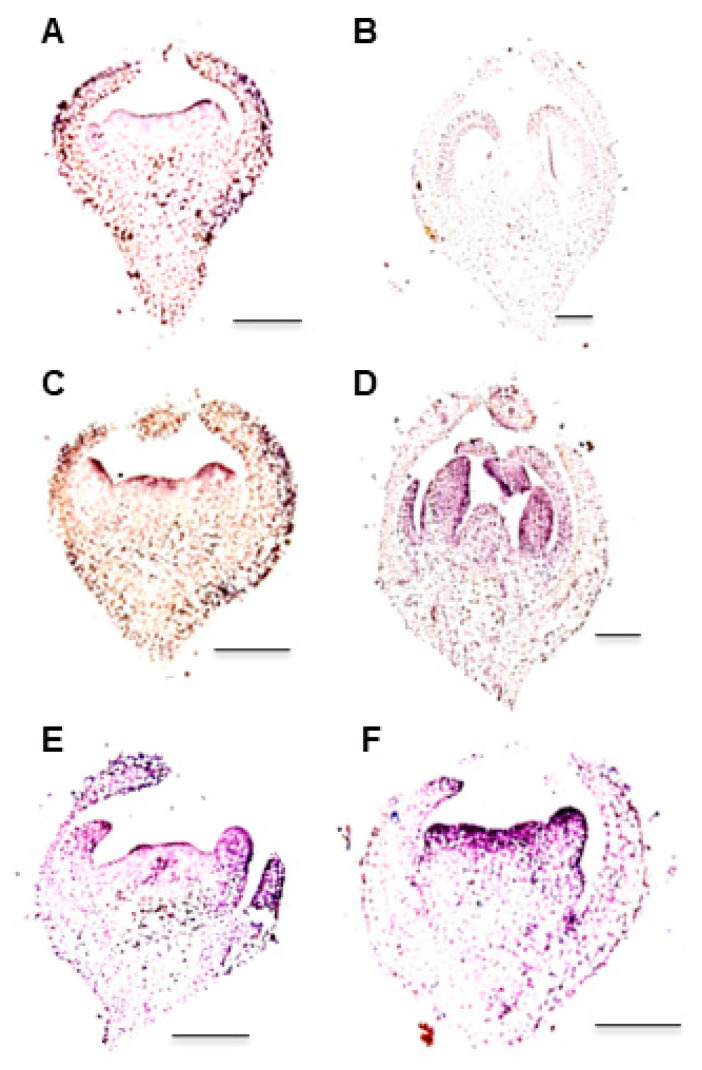
*FMF* expression in wild type flowers. (**A**–**D**) *FMF* expression wild type flowers revealed by in situ hybridization. (**A**,**B**) Flower sections were probed by *FMF* sense probe. (**C**,**D**) Flower sections were probed by *FMF* antisense probe. (**E**) *SlWUS* expression in wild type flowers. (**F**) *SlWUS* expression in *fmf* flowers. Scale bar, 100 μm.

## Data Availability

Materials generated in this study will be available upon request to the corresponding author (H.X.).

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
