# Peer review of "A SUPERMAN-like Gene Controls the Locule Number of Tomato Fruit"

_plants, 2023, doi:10.3390/plants12183341_

Round 1

Reviewer 1 Report

The manuscript by Znang et al. explores the genetic factors influencing flower development and fruit morphology in tomato. They identify a unique S. pimpinellifolium mutant, fmf, characterized by abnormal fruit. The authors examine fmf morphology, highlight an increased number of stamens and carpels compared to the wild type, and locate the gene responsible for this phenotype using next-generation sequencing. They identify a SUP homologue bearing a mutation linked to the fmf phenotype, and validate this finding by CRISPR-Cas9 and transcriptome analysis.

The study addresses an important research question in a thorough manner. The authors utilize an appropriate methodology, which is sufficiently detailed in the manuscript. The conclusions drawn are well-supported by the data presented. Overall, the manuscript is compelling, well-written, and interesting to read. I have only minor comments to add.

Minor:

line 2: I am somewhat confused by the title. Why "SUPMAN", is it a typo? 

line 38: "1" doesn’t seem necessary

lines 47-49: check the grammar, consider rewriting this sentence

line 119: "scanning electron microscopy"

line 139: check the scale bar units for typographic errors (100 mm? 500 mm?), the same comment for fig. 4, 6 and 7.

line 244: "as" is not necessary 

Author Response

Dear review editor,

On behalf of the authors, I would like express our gratefulness for your constructive comments and suggestions. We have corrected the errors you pointed, please see our point-by-point response below. All edits we made in the revision are highlighted in red.

Best wishes,

Han

-------------------------------------------------------------------------------

Point-by-point response to your comments:

line 2: I am somewhat confused by the title. Why "SUPMAN", is it a typo? 

Thank you for pointing out the typo error. It should read as “SUPERMAN”. Corrected.

line 38: "1" doesn’t seem necessary

removed

lines 47-49: check the grammar, consider rewriting this sentence

The sentence has been replaced.

line 119: "scanning electron microscopy"

corrected.

line 139: check the scale bar units for typographic errors (100 mm? 500 mm?), the same comment for fig. 4, 6 and 7.

These errors were introduced during reformatting the text font. Corrected.

line 244: "as" is not necessary 

Removed.

Reviewer 2 Report

REVIEW
of the paper SUPMAN-like gene controls locule number of tomato fruit by Mi Zhang, Enbai Zhou, Meng Li, Shenglan Tian, and Han Xiao
The paper addressees the genetic basis of the (micro)evolution of tomato. This is rather interesting paper, however, it should be published after some improvement.

Mainly, the work stands on the comparison of forms of the plants under a study.

Morphometry is rather well developed area of science, and botany, speci cally.
However, the Authors have said not a word on this issue. No rigorous statements, nor clear de nitions are provided. That is the core disadvantage of the paper.

However, I'm far from the idea to renovate completely the paper. The Authors
should extend the Discussion section with clear and comprehensive explanation of their avoidance to implement some standard tools and issues of the morphometry, say, analysis of topology of the patters of leaves and/or fruits, etc. Reciprocally, Introduction section references and at least brief description of those methods.

To summarize, I ought to say that this paper for sure could be published upon a
some renovation and improvement.

Author Response

Dear reviewer editor,

We agree with your point that morphometric tools can be used to effectively quantify leaf and fruit shape. There are also morphometric tools available, such as FlowerMorphology, ImageJ, Tomato Analyzer, MorphoGraphX-based 3D image processing and more. However, the phenotypes of flower and fruit development we focused in this study were qualitative, though we reported the changed numbers of floral organs caused in fmf mutants and corresponding wild type, which can be easily counted. Because we did not measure the variations in flower and fruit shape using any morphometric tools, we think it is not appropriate to mention and discuss irrelevant morphometric methods in the main text. We did not find the FMF gene plays any role in leaf development and no leaf data was collected and reported.

We do appreciate your suggestions and will try to apply morphometric tools in our future work as needed.